# Size/Age Models for Monitoring of the Pink Sea Fan *Eunicella verrucosa* (Cnidaria: Alcyonacea) and a Case Study Application

**Giovanni Chimienti** [1,2,*] **, Attilio Di Nisio** [3] **and Anna M.L. Lanzolla** [3]

1   Department of Biology, University of Bari Aldo Moro, Via Orabona 4, 70125 Bari, Italy
2   CoNISMa, Piazzale Flaminio 9, 00196 Roma, Italy
3   Department of Electrical & Information Engineering (DEI), Polytechnic University of Bari, 70125 Bari, Italy;
    attilio.dinisio@poliba.it (A.D.N.); anna.lanzolla@poliba.it (A.M.L.L.)
*   Correspondence: giovanni.chimienti@uniba.it; Tel.: +39-080-544-3344

**Abstract:** The pink sea fan *Eunicella verrucosa* is a habitat-forming octocoral living in the East Atlantic and in the Mediterranean Sea where, under proper circumstances, it can form large populations known as coral forests. Although these coral forests represent vulnerable marine ecosystems of great importance, these habitats are still poorly known, and their monitoring is almost non-existent to date. For this reason, we compared two different models to infer the age of *E. verrucosa* based on nondestructive measurements of the colonies' size, in order to highlight strengths and weaknesses of the existing tools for a potential application in long-term monitoring. We also applied the two models on a case-study population recently found in the northwest Mediterranean Sea. Our results showed which model was more reliable from a biological point of view, considering both its structure and the results obtained on the case study. However, this model uses solely the height of the colonies as proxy to infer the age, while the total branch fan surface area could represent a more appropriate biometric parameter to monitor the size and the growth of *E. verrucosa*.

**Keywords:** Anthozoa; gorgonian; corals; coral habitat; coral forest; VME; mesophotic; modeling; biometry; Mediterranean

---

## 1. Introduction

The pink sea fan *Eunicella verrucosa* (Pallas, 1766) is an octocoral species (order Alcyonacea, family Gorgoniidae) living in the East Atlantic, from Ireland to Angola, and in the Mediterranean Sea [1–5]. This species can be found at a depth of 2 to 60 m in the Atlantic Ocean, while it can reach a depth of 200 m in the Mediterranean Sea [1,4–7]. Under proper oceanographic conditions including strong currents and temperate/cold waters, *E. verrucosa* can form large colony aggregations known as coral forests, settling on hard bottoms sometimes covered by a thin sediment veneer [5]. Similar to other animal forests (*sensu* [8]), octocoral assemblages provide biomass, structural complexity and aesthetic value to coastal communities, sustaining a rich biodiversity [5,9–17]. Coral forests, comprised of *E. verrucosa* represent a Vulnerable Marine Ecosystem (VME) because of their vulnerability to human pressures, according to the Food and Agricultural Organization (FAO) [18,19]. This vulnerability is based on the rarity, the functional significance, the fragility (both physical and functional), and the structural complexity of the coral forest, as well as the species' life-history traits (e.g., slow growth rate, late age of maturity, low or unpredictable recruitment, and extended life expectancy) that makes recovery difficult after a fishing impact. Indeed, bottom-contact fishing gears such as trawl nets, dredges, longlines, and artisanal fishing nets (e.g., gillnets and trammel nets) can have both direct and

indirect impacts on these coral forests, representing one of the main anthropogenic threats together with pollution and climate change [5,20–26]. Furthermore, the stranding of *E. verrucosa* has been recently documented on the southwest coast of England, where the coral appeared to be entangled in lost fishing gears, as well as in domestic marine litter [23].

From the 1960s to the 1970s, colonies of *E. verrucosa* were collected as souvenirs in the northeast Atlantic, where the species has been currently protected from intentional damage [27]. It represents a major heritage species [28,29] and it is listed as "vulnerable" on the Red List of Threatened Species by the International Union for Conservation of Nature and Natural Resources (IUCN) [30]. More locally, the status of *E. verrucosa* in the Mediterranean Sea seems to be slightly better than elsewhere, as it is listed as "near threatened" according to IUCN [31]. In fact, records of *E. verrucosa* are increasing in this basin thanks to the implementation of deep diving techniques, the carrying of Remotely Operated Vehicle (ROV) explorations, as well as the application of a citizen science approach involving scuba divers and fishermen [5–7,25]. The emerging picture is that of a species widespread in the Mediterranean Sea, although large populations constituting coral forests are still poorly known [5]. Due to its vulnerability and ecological importance, *E. verrucosa* is receiving increasing attention and, similar to other forest-forming corals, further studies are likely to unveil new populations of this species in the mesophotic zone. Together with the finding of new coral forests aiming to assess the overall distribution of these VMEs, their monitoring is essential to understand ongoing and future changes (both natural and anthropogenic) [32].

Underwater observations and size measurements carried out on living colonies have allowed the development of two almost contemporary and independent models of growth and size/age relationship for *E. verrucosa*, one in the Atlantic Ocean [33] and one in the Mediterranean Sea [6]. In this study we compared the two existing models to infer the age of *E. verrucosa* based on the size of the colonies, in order to analyze pros and cons for long-term monitoring. Moreover, we applied the two models on a recently described forest of *E. verrucosa* in the northeast Mediterranean Sea [5] as a case study. New indications to improve age modeling and to obtain more reliable data on *E. verrucosa* populations are also provided.

## 2. Materials and Methods

### 2.1. Models Considered

We considered two different models for the correlation of size and age in *E. verrucosa*. Model A [33] was developed based on 70 colonies tagged and surveyed between 2006 and 2008 in the Gulf of Morbihan (northwest Atlantic coast of France), at a depth of 8–20 m. The authors measured the colonies with in situ photographic recordings, and then used imaging analysis to estimate the total Branch Fan Surface Area (BFSA, cm$^2$), namely, the measure of the surface that is occupied by the organism on photographs in two-dimensional (2D) projection (Figure 1). The BFSA was calculated considering both height and width, and linearly correlated to the BFSA with different coefficients (Figure 2a,b). A correction factor for branch overlapping was introduced considering that *E. verrucosa* colonies could exhibit a three-dimensional (3D) morphology. Individual growth was assessed by measuring the increase in the BFSA of tagged colonies during the three years of study, and the following model that correlated the age of the colony with the BFSA was proposed (Figure 2a) [33]:

$$BFSA = 1.25 \times age^{1.73}. \tag{1}$$

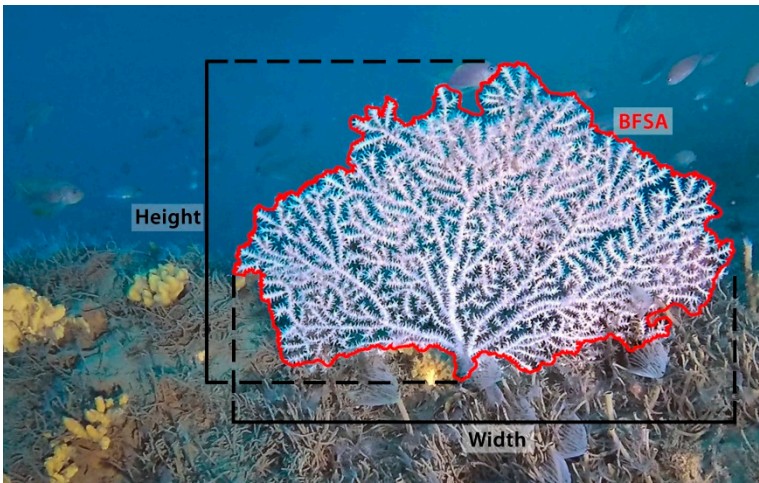

**Figure 1.** Colony of *Eunicella verrucosa* with indication of height, width, and branch fan surface area (BFSA).

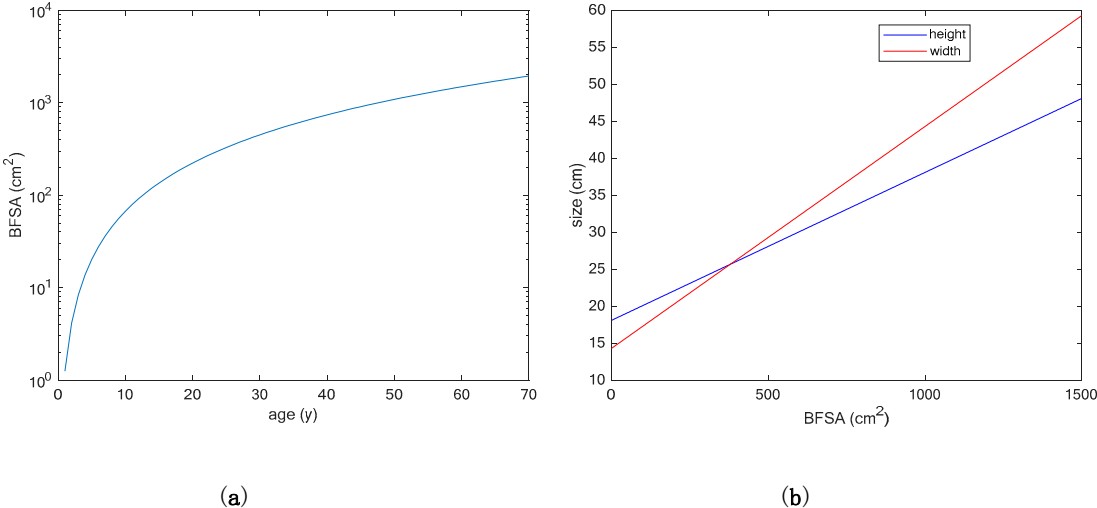

**Figure 2.** Model A [33]. (**a**) Estimated behavior of the BFSA (branch fan surface area) versus age in *Eunicella verrucosa*; (**b**) Linear correlation between the BFSA and height (blue) or width (red) of the colonies.

The model was based on colonies having an initial BFSA between approximately 3 and 50 cm$^2$. The latter size corresponded to an estimated age of about 30 years and an estimated height of 28 cm.

The same study reported the following linear relations between BFSA and height (*h*) and width (*w*) (Figure 2b), that have been considered in this study:

$$h = 0.02 \times BFSA + 18.08, \tag{2}$$

$$w = 0.03 \times BFSA + 14.29. \tag{3}$$

Model B [6] was developed based on 15 colonies surveyed every two years between 1997 and 2006 in the Riou archipelago (Mediterranean coast of France), at a depth between 20 and 40 m. The authors measured in situ the maximum height of the colonies (between 5 and 40 cm) using a decimetre with a millimetre scale. Individual growth rate was assessed by measuring the increase in the height of tagged colonies during the nine years of study, and the following Von Bertalanffy growth function that correlated height (*h*) and age of the colonies was proposed (Figure 3) [6]:

$$h = 17.94 \times ln(age) - 18.39. \tag{4}$$

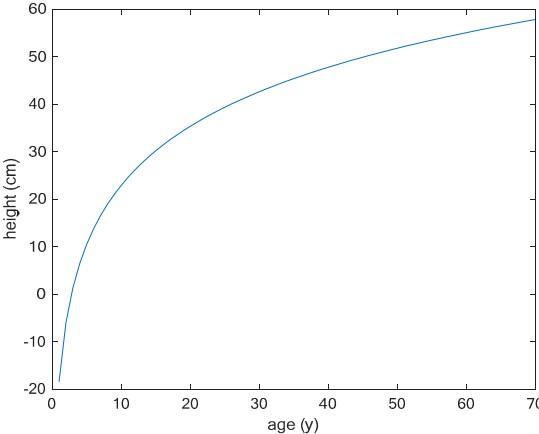

**Figure 3.** Height estimation of the colonies as a function of age in *Eunicella verrucosa* according to model B [6].

For both models, the behavior of height as a function of age was analyzed to compare the performance of the two models. In particular, the following relationship was obtained from the modeling proposed in [33]:

$$h = 18.08 + 2.5 \times age^{1.7}. \tag{5}$$

### 2.2. Study Site

The extensive population of *E. verrucosa* recently found off the city of Sanremo (Ligurian Sea, northwest Mediterranean) [5] was considered as a case study. High-resolution photos of *E. verrucosa* colonies with a size reference (resolution of 1 cm) were collected by technical scuba dives in order to estimate height and width of the colonies. A total of 82 colonies, found at a depth of 65–70 m, were photographed. Height and width values were estimated for each colony by means of image analysis with a raster graphics editor.

The empirical cumulative distribution functions of height and width in the observed population were calculated and compared. Skewness and kurtosis, calculated for height and width, as well as for age, were not corrected for bias since no assumptions on distributions were made.

The relation between width *w* and height *h* for the population of Sanremo was analyzed by calculating the linear regression of width on height for two models, i.e., $w = a \times h + b$ and $w = a \times h$. The standard error of regression coefficients, their 95% confidence intervals, and *p*-values were evaluated by means of a *t*-statistic. The linear regression was compared with a scatter plot for the population off Sanremo and with a regression illustrated in [33] relevant to model A.

The prediction of age from height for the population off Sanremo was calculated according to models A and B. For *h* = 18.08 cm, age was zero according to model A. Hence, the result of that model for colonies below 18.08 cm was considered to be zero years (indicating colonies younger than one year). Age distribution according to the two models was analyzed by means of histograms with five year bin widths, whose values were normalized to obtain a probability density function (i.e., relative frequencies were divided by bin width). Histograms were smoothed by fitting a Gaussian kernel with a bandwidth of 5 years, and the enforcement of positive age values were obtained by means of the reflection method implemented by the *ksdensity* MATLAB function.

## 3. Results

### 3.1. Comparing Two Different Models

Models A and B have a very different structure. The comparison of the height as a function of colonies' age shows that model A is characterized by an exponential increase of height with age (Figure 4), a feature that is poorly representative from a biological point of view. On the contrary,

model B seems to better represent the corals' trend of growth, with small colonies having a first stage of fast growth and a flattening curve with increasing age (Figure 4).

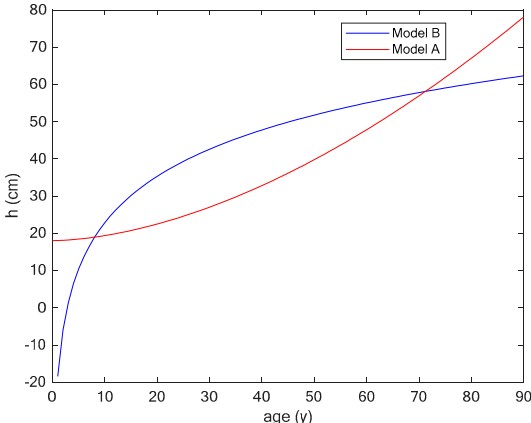

**Figure 4.** Comparison of height estimation as a function of age in *Eunicella verrucosa* by using model A and B.

## 3.2. Case Study

The forest of *E. verrucosa* recently found off the city of Sanremo, in the Ligurian Sea, represented a good case study to test the two models due to the presence of both small and large colonies [5].

The empirical distribution function of the colonies' size showed that most of the population had similar values of height and width, while large-size colonies tended to be wider than high (Figure 5a). On the contrary, colonies of smaller size (likely to be juveniles) were generally higher than wide, considering that 12% of the colonies was less than 10 cm wide, while only 6% of the colonies was less than 10 cm high (Figure 5b). In fact, young colonies are usually unbranched or with very short branches, while they generally develop a fan shape in later years. This is in accordance with the model proposed by Coz et al. [33] in the North Atlantic.

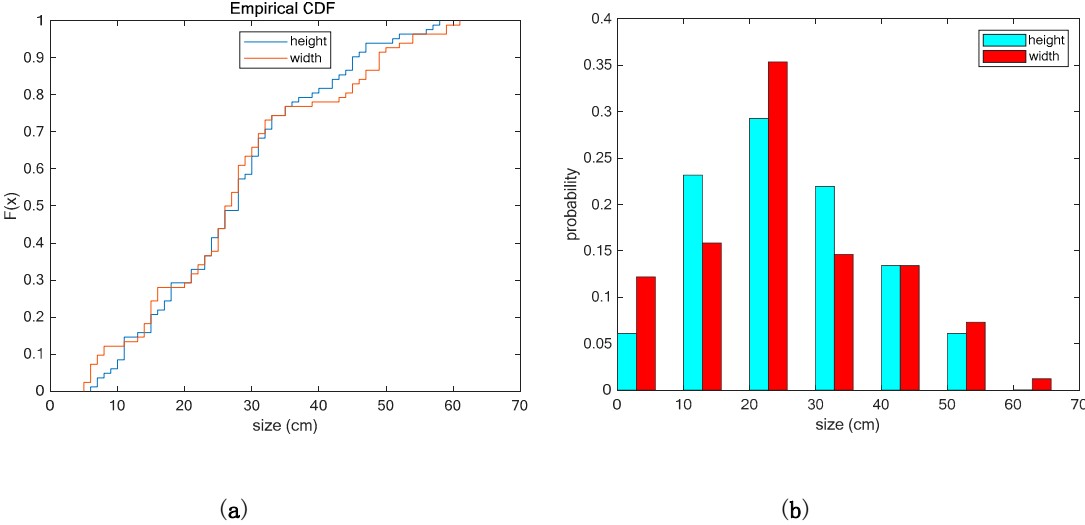

(a)                                    (b)

**Figure 5.** (**a**) Empirical distribution functions of height and width of the *Eunicella verrucosa* colonies off Sanremo; (**b**) Histogram of height and width for classes 10 cm wide.

Maximum height and width were 58 and 61 cm, respectively. The size-frequency distribution was mesokurtic, with a slight positive skewness (Table 1).

**Table 1.** Size statistics for the *Eunicella verrucosa* population monitored off Sanremo.

|  | Mean (cm) | Median (cm) | Standard Deviation (cm²) | Skewness (cm³/cm³) | Kurtosis (cm⁴/cm⁴) |
|---|---|---|---|---|---|
| Height | 27.6 | 28.0 | 12.8 | 0.4 | 2.6 [1] |
| Width | 27.9 | 26.5 | 14.6 | 0.4 | 2.4 |

[1] This value is wrongly reported as −0.4 in [5].

The test of the linear regression of width $w$ on height $h$ ($w = a \times h + b$) on the population off Sanremo showed a slope value of $a = 1.05$, with a standard error of 5% and a 95% confidence interval equal to 0.95–1.14. The obtained intercept was $b = -1.0$ cm. The 95% confidence interval was large, (i.e., −4.0, 1.9 cm), with $p$-value = 0.5 for the $t$-statistic of the hypothesis test that the coefficient is null. Considering the obtained results, the data were fitted by means of a linear regression without intercept, as illustrated below. The latter model is also more significant from a biological point of view, considering that at time zero, when the coral larva settles down, the size of the colony is null.

The regression of width on height ($w = a \times h$) for the measured colonies was characterized by a slope $a = 1.02$ with a standard error of 2% and a 95% confidence interval equal to 0.98–1.06 (Figure 6). The linear relation between width and height relevant to model A, obtained from [33], is characterized by a slope of 1.5 (Figure 6). Considering the scatter plot of width and height (Figure 6), it is evident that each colony tended to be more developed in width at large sizes (records mostly above the quadrant bisector), while small colonies were higher than wide. This confirmed, at the level of each colony (Figure 6), what was observed with the empirical distribution functions (Figure 5a) where, instead, width and height were treated independently.

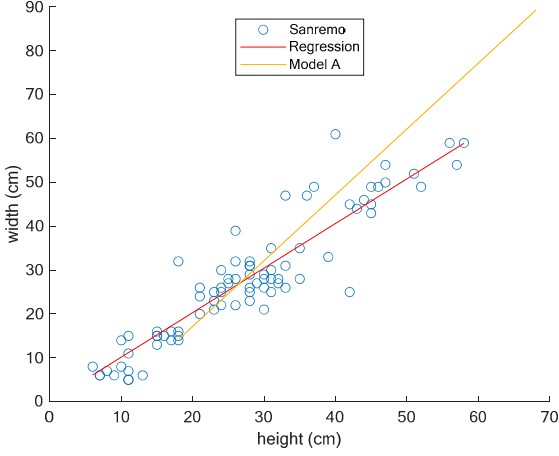

**Figure 6.** Height and width of the *Eunicella verrucosa* colonies off Sanremo, with linear regression (red line). In yellow the regression from model A [33].

Application of the Two Models

The age of the *E. verrucosa* colonies monitored off Sanremo, calculated on the basis of the height values, showed different results between model A and model B (Figure 7). In particular, model A is not valid when height is less than 18 cm, since in that case (5) cannot be solved. This happened for 29% of the colonies, which were, therefore, estimated to have an age of zero years (i.e., younger than one year, see left side of Figure 7). Although all the size classes from 6 to 58 cm were present in the studied population (Figure 5b), the prediction from model A jumped from several colonies younger than one year to a colony 16 years old (Figure 8a), proving to be scarcely effective for small sizes. On the contrary, model B showed a gradient of predicted age from four to nine years old for the same small colonies, although the smallest colony (6 cm in height) could be likely to be younger than four years old.

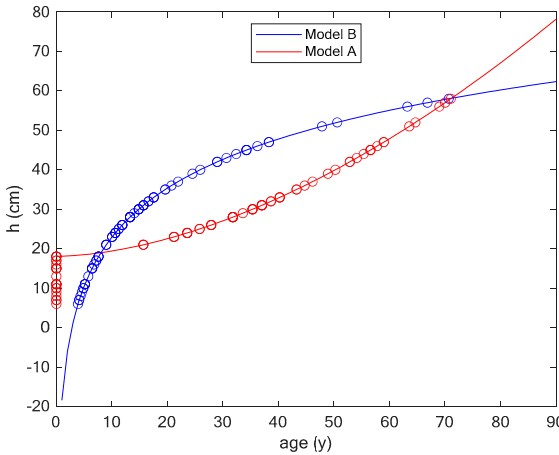

**Figure 7.** Age of the *Eunicella verrucosa* colonies off Sanremo, obtained from the height measure, using the two considered models.

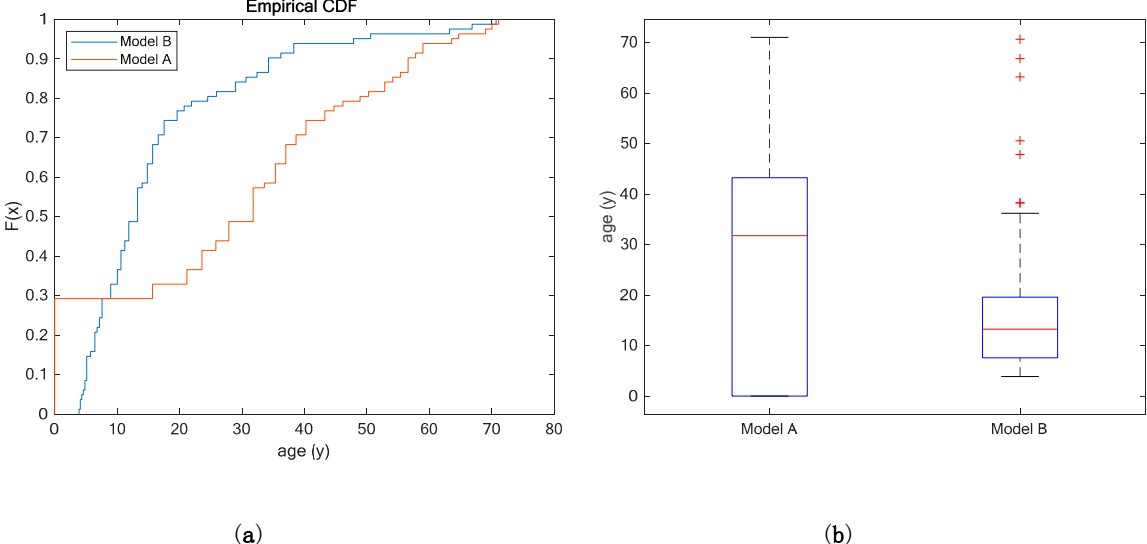

(**a**)          (**b**)

**Figure 8.** Age of the *Eunicella verrucosa* colonies off Sanremo estimated using model A and B. (**a**) Empirical distribution function; (**b**) Box plot showing median (red), first and third quartile (blue), range (whiskers), and outliers (red crosses). Here, outliers are values at a distance above the third quartile or below the first quartile greater than 1.5 times the interquartile range.

With an increasing size, the two models diverged considerably, with model B being more conservative than model A (Figure 8b). On the highest size found, i.e., 58 cm, the models showed almost the same estimated age (approximately 70 years old). This is likely to be close to the maximum height that *E. verrucosa* can reach, according to present knowledge, and is the point of contact between the two curves (Figure 7). Larger sizes could eventually show discordance between the two models, considering that model A predicts a younger age as compared with model B for heights above 58 cm (Figure 7).

The largest colonies were of a few large, old ones that were included in the boxplot of model A (Figure 8b), while they were present as outliers for model B. In the latter, the interquartile range was smaller as compared with model A, highlighting the higher conservativism of model B.

The predicted age-frequency distributions are shown in Figure 9a,b as histograms and as fittings with a Gaussian kernel. It further highlighted the presence of a gap for small sizes in model A (Figure 9a), and a distribution of the data that seemed to suggest that model B was more reliable from a biological point of view. In fact, the latter showed the presence of a highly skewed and leptokurtic

distribution (Figure 9b and Table 2), with many colonies younger than 20 years old and a long right tail of older colonies represented by those broadly higher than 35 cm.

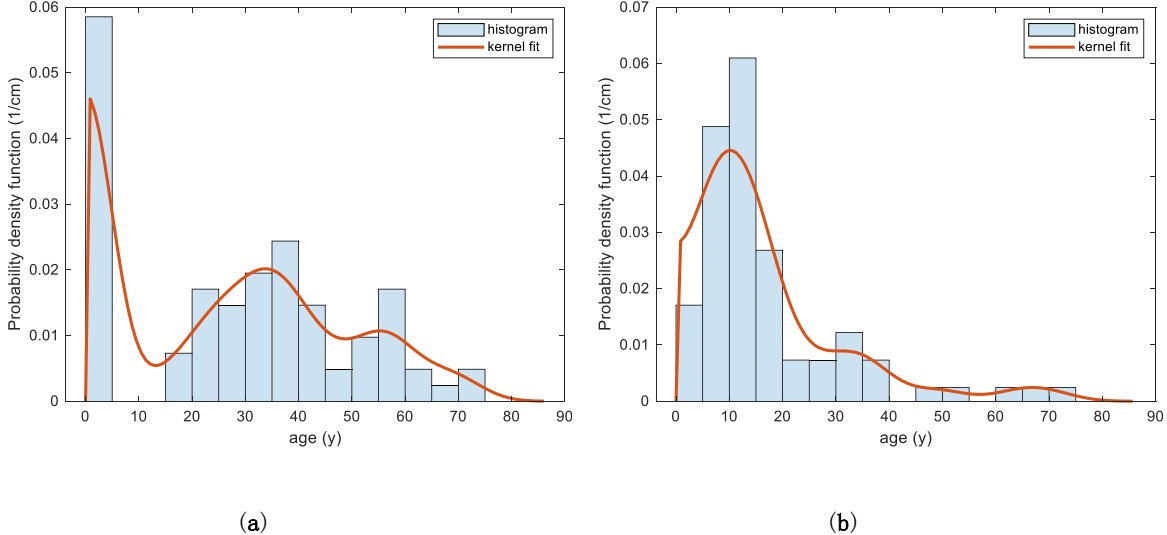

**Figure 9.** Age-frequency distribution of the *Eunicella verrucosa* population monitored off Sanremo, by model. (**a**) Model A; (**b**) Model B. The histograms have been normalized to become comparable with the estimated probability density function. The bin width of the histogram and the bandwidth of the Gaussian kernel are both set to 5 years.

**Table 2.** Age statistics for the *Eunicella verrucosa* population monitored off Sanremo, y = years.

| | Mean (y) | Median (y) | Standard Deviation (y²) | Skewness (y³/y³) | Kurtosis (y⁴/y⁴) |
|---|---|---|---|---|---|
| Model A | 27.9 | 31.8 | 21.9 | 0.1 | 1.9 |
| Model B | 17.0 | 13.3 | 14.3 | 2.0 | 6.8 [1] |

[1] This value is wrongly reported as 4.2 in [5].

## 4. Discussion

The two size/age models available, thus far, for monitoring of *E. verrucosa* [6,33] were developed in two different areas, i.e., the northwest Mediterranean Sea and the northeast Atlantic Ocean. These areas are characterized by different environmental conditions (e.g., temperature, water currents, and depth of occurrence) which affect trophism and coral physiology, influencing the growth rate of *E. verrucosa* [6,33]. Thus, the difference between the results of the two models observed in this study could be partly due to the different areas of application. However, a standardized method to estimate the age of the colonies based on their size should be adjusted also considering the study area, at least at the basin level (e.g., introducing a different correction factor for the Atlantic Ocean and the Mediterranean Sea populations). Model B seemed to be stronger than model A, considering both the structure of the formula (Figure 4) and the results of the case study (Figure 7). This is related to the fact that, independently from the study area, the relationship between size and age in animals, including corals, follows a von Bertalanffy curve where young stages grow very fast with a decrease in growth rate according to age [34–36]. Notwithstanding, the height of the colonies used in model B can be limiting, considering the importance of width measures in *E. verrucosa*, at least for the larger colonies (Figure 6). The importance of width measures has also been highlighted in other congeneric species, such as *E. singularis* [33,37]. In fact, modular colonial organisms such as *E. verrucosa* and other sea fans form colonies by polyp iteration, and therefore the total branch fan surface area (BFSA), which is a function of both height and width, is probably the most appropriate estimate of its growth and the more suitable measure to be linked to the age [33,38].

The greatest limitation of both model A and B is that the predicted age of the colonies is based on indirect information rather than direct measures. For this reason, it is essential to improve these models or to identify a new one based on the empirical measure of the colonies' age using geochemical dating methods (e.g., radiocarbon), already adopted for the ageing of other coral species [39–41], as well as growth ring analysis, as carried out on other *Eunicella* species, although with some limitations due to poorly differentiated rings at the periphery of the skeleton of large colonies [42,43]. These methods are based on the analysis of the basal cross-section of the colony, being unsustainable for long-term monitoring surveys, as well as incompatible for ethical and conservation reasons, considering that *E. verrucosa* represents a vulnerable species [30].

This important VME indicator taxa (*sensu* [19]) is affected by bottom-contact fishing gears and it might occur as bycatch during fishing operations. In fact, the exclusion of demersal towed equipment has benefited *E. verrucosa* populations and other benthic taxa in the Lyme Bay Marine Protected Area (MPA; southwest England) [44], emphasizing the role of MPAs for the conservation of vulnerable and threatened species. However, bottom longlines and artisanal fishing nets (e.g., trammel nets and gillnets), often considered of scarce impact on benthic communities, also have a relevant bycatch rate on mesophotic and deep-sea coral species [24,45,46]. These gears can have a significant impact on *E. verrucosa*, as they can bycatch the colonies during fishing operations, but also lost fishing gears can be entangled in the colonies and damage them (ghost fishing) [5,23]. The exploitation of fishing data can allow for the identification and recording of vulnerable coral communities [47–51]. The involvement of fishermen could help to keep these accidentally collected colonies, in order to maximize the bycatch and provide specimens for invasive analysis aiming to understand the longevity and the growth rate of this species. These data could also be used to evaluate the impact of fishing gear on this vulnerable species, while non-destructive survey methods (i.e., ROV and technical diving) should be incentivized to identify and to monitor coral populations as a valid alternative to destructive sampling.

The possibility to directly measure the age of *E. verrucosa* colonies based on samples represents an opportunity to correlate size and age in this species, in order to build a stronger model. Moreover, the finding of a rich *E. verrucosa* forest would represent a population in the Atlantic Ocean that could provide a better comparison with the Mediterranean ones studied thus far. Such information is essential to the development of a standardized and non-invasive method for long-term monitoring of *E. verrucosa* and its population structure. This could lead to visual methods for identification, quantification, and eventually, measurement of the colonies. For instance, visible, multispectral, or hyperspectral imaging could be used, as already has been carried out for other coral species [52–55], as well as in other fields of research, from the study of animals and plants [56–58] to marine biogeochemistry [59].

**Author Contributions:** Conceptualization, G.C., A.D.N., and A.M.L.L.; methodology, G.C., A.D.N., and A.M.L.L.; software, A.D.N. and A.M.L.L.; validation, G.C.; formal analysis, A.D.N. and A.M.L.L.; investigation, G.C.; resources, G.C., A.D.N., and A.M.L.L.; data curation, G.C.; writing—original draft preparation, G.C.; writing—review and editing, G.C., A.D.N., and A.M.L.L.; visualization, G.C., A.D.N., and A.M.L.L.; supervision, G.C., A.D.N., and A.M.L.L.; project administration, G.C.; funding acquisition, G.C. All authors have read and agreed to the published version of the manuscript.

**Funding:** This research was funded by the Italian Ministry of Education, University and Research (Ministero dell'Istruzione, dell'Università e della Ricerca; Programma Operativo Nazionale, PON 2014–2020), grant AIM 1807508-1, Linea 1, and by the Italian Ministry for Environment, Land and Sea Protection (Ministero dell'Ambiente e della Tutela del Territorio e del Mare), as part of the Italian monitoring program for the implementation of the Marine Strategy Framework Directive (European Union, 2008/56/EC).

**Acknowledgments:** The authors wish to thank Maurizio Delfini for the support with underwater videos and photos.

**Conflicts of Interest:** The authors declare no conflict of interest. The funders had no role in the design of the study; in the collection, analyses, or interpretation of data; in the writing of the manuscript, or in the decision to publish the results.

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
