# Peer review of "Size/Age Models for Monitoring of the Pink Sea Fan Eunicella verrucosa (Cnidaria: Alcyonacea) and a Case Study Application"

_jmse, doi:10.3390/jmse8110951_

Round 1
Reviewer 1 Report
The paper deals with an interesting topic, considering the role that the studied species play in forming coral forests; the paper is well written and designed, and in my opinion, it deserves publication in the present form; I suggest only a few tips and corrections, listed below:
lines 19, 67: NE Mediterranean Sea; I think, it stands for NW, Northwest, because the authors used, as a case study, a population of The Ligurian Sea.
Please check throughout the text
Line 237. In this context, discussing the impacts of bottom trawling on these vulnerable habitats, I suggest the authors quote this paper: "Sea pens in the Mediterranean Sea: Habitat suitability and opportunities for ecosystem recovery DOI:10.1093/icesjms/fsy010"
Line 240. "gear" here, with this meaning, (equipment) is uncountable.
Author Response
We thank you the reviewer for the useful suggestions. Please find below a point-by-point reply.
lines 19, 67: NE Mediterranean Sea; I think, it stands for NW, Northwest, because the authors used, as a case study, a population of The Ligurian Sea.
Please check throughout the text
Reply: We changed it. Thanks for highlighting this typo.
Line 237. In this context, discussing the impacts of bottom trawling on these vulnerable habitats, I suggest the authors quote this paper: "Sea pens in the Mediterranean Sea: Habitat suitability and opportunities for ecosystem recovery DOI:10.1093/icesjms/fsy010"
Reply: We agree. Quotation added at line 246.
Line 240. "gear" here, with this meaning, (equipment) is uncountable.
Reply: we used the word equipment as suggested, in order to reduce the use of the word “gear”.
Reviewer 2 Report
Dear Authors,
Manuscript jmse-995400 'Size/age models for the monitoring of the pink sea fan Eunicella verrucosa (Cnidaria: Alcyonacea) and case study application' by Chimienti, Di Nisio, Lanzolla has been revised. This research is based on an original idea and the results could be useful to improve knowledge and therefore protection of vulnerable, long-living benthic species. In my opinion, the manuscript is worth of publication with minor revision.
Main comment is that some paragraphs could be better structured to emphasize the aim of the work and its results. Specific comments are listed below.
Abstract
Please, change ‘vulnerable marine ecosystems’ in ‘Vulnerable Marine Ecosystems (VME)’
Introduction
Line 33. Do you have any information (from literature or based on your experience) about the arrangement of the colonies respect the substrate? May the colony’s orientation influence the vulnerability of the species?
Lines 50-52. Please, explain why the status of E. verrucosa in the Mediterranean Sea is slightly better than elsewhere.
Lines 52-54. If possible, supply percentage of records of E. verrucosa from fisheries respect to records obtained with non-destructive methods. It would be interesting to review citizen science records of E. verrucosa from the Mediterranean and to compare them with generic results obtained by Krželj et al. 2020, Fig. 4. https://doi.org/10.3390/d12080311
Line 61-62. Besides the two papers about E. verrucosa, are there other models to evaluate the age of gorgonians? I suggest summarising here methods applied to determine the age of most common Mediterranean gorgonians and explaining why it is so important to estimate the age of these long-living species.
Lines 62-68. Please, highlight benefits emerging from your results in order to make your research of broader interest. I think that first aim of your work is to supply new indications to improve age modelling and to obtain more reliable data on habitat-forming corals. E. verrucosa is a case study, but your results might be generalised.
Methods
Please, subdivide this session in two sub-paragraphs, one about the comparison between models A and B and one inherent to your study.
Discussions
Line 216. ‘standardized method to estimate the age of the colonies based on the size should be adjusted also considering the study area, at least at basin level’ How do you want to adjust the method? (You explained this ahead in the manuscript; however, you should better organize paragraphs to avoid fragmentation).
Line 239. Please, emphasize the role of Marine Protected Areas for conservation of these vulnerable species.
Line 243. ‘The exploitation of fishing data, indeed, can allow to identify vulnerable coral communities’ These data should be used above all to evaluate the impact of destructive fishing gears on the vulnerable species. Moreover, you should state that non-destructive survey methods (i. e., R.O.V., technical diving etc.) should be incentivized to identify and to monitor populations of gorgonians in order to limit fishing activities in those areas. E. verrucosa shows a long lifespan and increases the topography complexity; therefore, the rarefaction of this species could lead to the lost of its ecosystem services.
Figure 1. ‘Colony of Eunicella verrucosa with indication of height, width and branch fan surface area (BFSA)’ Add ‘according to model A [33]
Author Response
We thank you the reviewer for the useful suggestions. Please find below a point-by-point reply.
Abstract
Please, change ‘vulnerable marine ecosystems’ in ‘Vulnerable Marine Ecosystems (VME)’
Reply: Done.
Introduction
Line 33. Do you have any information (from literature or based on your experience) about the arrangement of the colonies respect the substrate? May the colony’s orientation influence the vulnerability of the species?
Unfortunately we have no data about it, although the orientation of the colonies should be driven by the main currents, as it happens in many other gorgonians characterized by a fan shape. However, we don’t think that this could influence the vulnerability of the species, that would be vulnerable with whatever orientation.
Lines 50-52. Please, explain why the status of E. verrucosa in the Mediterranean Sea is slightly better than elsewhere.
Reply: it has been classified as ‘near threatened’ by IUCN. The reason why the status of the species is better in the Mediterranean compared to the Atlantic is still not clear. It could be related to a more efficient protection, to a sort of ‘climatic refuge’ or just to larger scientific efforts in understanding the distribution of the species. However this is just our idea that cannot be supported by data or literature at the moment, thus we prefer not to include this part in the text.
We changed ‘is’ with ‘seems to be’.
Lines 52-54. If possible, supply percentage of records of E. verrucosa from fisheries respect to records obtained with non-destructive methods. It would be interesting to review citizen science records of E. verrucosa from the Mediterranean and to compare them with generic results obtained by Krželj et al. 2020, Fig. 4. https://doi.org/10.3390/d12080311
Reply: This information is taken from the literature cited and, unfortunately, we do not currently have quantitative data. Moreover, this comparison is very interesting but it goes beyond the scope of the manuscript.
Line 61-62. Besides the two papers about E. verrucosa, are there other models to evaluate the age of gorgonians? I suggest summarizing here methods applied to determine the age of most common Mediterranean gorgonians and explaining why it is so important to estimate the age of these long-living species.
Reply: There are other models developed on other species. However, we prefer to keep the focus on E. verrucosa rather than summarize methods applied to other species thus far. In fact, other Mediterranean octocorals such as Paramuricea clavata, Eunicella cavolini and the precious red coral Corallium rubrum have been more extensively studied in terms of growth rate and age, often with different methods and different results. Including this in the introduction (and eventually in the discussion) in a proper way will lengthen the text and go beyond the scope of this manuscript. By the way, we are grateful for this comment because it could allow a review of all the methods available on octocorals in the future.
Lines 62-68. Please, highlight benefits emerging from your results in order to make your research of broader interest. I think that first aim of your work is to supply new indications to improve age modelling and to obtain more reliable data on habitat-forming corals. E. verrucosa is a case study, but your results might be generalised.
Reply: We agree, and we added the sentence suggested.
Methods
Please, subdivide this session in two sub-paragraphs, one about the comparison between models A and B and one inherent to your study.
Reply: We subdivided into section “2.1 Models considered” and “2.2 Study site”.
Discussions
Line 216. ‘standardized method to estimate the age of the colonies based on the size should be adjusted also considering the study area, at least at basin level’ How do you want to adjust the method? (You explained this ahead in the manuscript; however, you should better organize paragraphs to avoid fragmentation).
Reply: we rephrased accordingly.
Line 239. Please, emphasize the role of Marine Protected Areas for conservation of these vulnerable species.
Reply: Done.
Line 243. ‘The exploitation of fishing data, indeed, can allow to identify vulnerable coral communities’ These data should be used above all to evaluate the impact of destructive fishing gears on the vulnerable species. Moreover, you should state that non-destructive survey methods (i. e., R.O.V., technical diving etc.) should be incentivized to identify and to monitor populations of gorgonians in order to limit fishing activities in those areas. E. verrucosa shows a long lifespan and increases the topography complexity; therefore, the rarefaction of this species could lead to the lost of its ecosystem services.
Reply: We agree, and we corrected as follow: “In fact, the exclusion of demersal towed equipment has benefited E. verrucosa populations and other benthic taxa in the Lyme Bay Marine Protected Area (MPA; SW England) [44], emphasizing the role of MPAs for the conservation of vulnerable and threatened species.”.
Figure 1. ‘Colony of Eunicella verrucosa with indication of height, width and branch fan surface area (BFSA)’ Add ‘according to model A [33]
Reply: We didn’t quote the reference because this figure shows what is generally intended as BFSA, in [33] but also in our paper.
Reviewer 3 Report
Some issues surround the MS with English. One factual error in introduction:
Line 15 - non-existent, not inexistent
Line 29 - the northern limit is not Scotland, but Ireland.
Line 35 - 'comprised of', Not 'like'
Line 36 - delete 'ones'
Line 70-71 - re-write - poor English.#
Line 78-81 - Figure 2a is based on modelled data from age of colonies related to 3 years of growth estimates. It's important here to show this data. Were the growth estimates of colonies at different size ranges? This is very important, as the conclusions of the paper are drawing from rapid growth in early years (0-16 years), and slow growth thereafter. So we (the reader) needs to be made aware of the populations, and their ranges from which the subsequent data is drawn. And how that data then is used to illustrate / match the field observations from this study. A similar criticism can be aimed at the assessment of size/age curve in Figure 3, although the study was undertaken over more years, but was of fewer colonies - so will be more accurate assuming that colonies at different sizes were selected, and that there is a consistency of morphometric change over time. From what original sizes of colonies were age assessments made, and with how many colonies at each age 'bin'?
Line 172-173 - re-write.
Line 243-244 - re-write.
Line 251 - remove 'assessement'
Lines 251-255 - poor English. And too long. This should be 2 sentences.
Author Response
We thank you the Reviewer for the useful suggestions. Please find below a point-by-point resply.
Line 15 - non-existent, not inexistent
Reply: Done.
Line 29 - the northern limit is not Scotland, but Ireland.
Reply: We have based our statement on Grasshoff 1(992), where the species is reported up to Scotland.
Line 35 - 'comprised of', Not 'like'
Reply: Done.
Line 36 - delete 'ones'
Reply: Done.
Line 70-71 - re-write - poor English.#
Reply: We rewrote as follow: “We considered two different models for the correlation of size and age in E. verrucosa”
Line 78-81 - Figure 2a is based on modelled data from age of colonies related to 3 years of growth estimates. It's important here to show this data. Were the growth estimates of colonies at different size ranges? This is very important, as the conclusions of the paper are drawing from rapid growth in early years (0-16 years), and slow growth thereafter. So we (the reader) needs to be made aware of the populations, and their ranges from which the subsequent data is drawn. And how that data then is used to illustrate / match the field observations from this study. A similar criticism can be aimed at the assessment of size/age curve in Figure 3, although the study was undertaken over more years, but was of fewer colonies - so will be more accurate assuming that colonies at different sizes were selected, and that there is a consistency of morphometric change over time. From what original sizes of colonies were age assessments made, and with how many colonies at each age 'bin'?
Reply: We have now added in methods all the information available, such as the size range of the colonies and more parameters used for the models. Further details, such as how many colonies at each age 'bin', are not available as data in the published paper and we were not able to use them also in our analysis. However, this information is reported in the graphs published by the authors and we quote the relevant paper, so the reader can go and check it for more details.
Line 172-173 - re-write.
Reply: We rewrote as follow: “The age of the E. verrucosa colonies monitored off Sanremo, calculated on the basis of the height values, showed different results between model A and model B”
Line 243-244 - re-write.
Reply: We rewrote as follow: “These gears can have a significant impact on E. verrucosa, as they can detach and bycatch the colonies during fishing operations, but also lost fishing gears can be entangled in the colonies and damage them (ghost fishing)”
Line 251 - remove 'assessement'
Reply: Done.
Lines 251-255 - poor English. And too long. This should be 2 sentences.
Reply: We rewrote as follow: “The possibility to directly measure the age of E. verrucosa colonies based on samples represent a need step to correlate size and age in this species, in order to build a stronger model. Moreover, the finding of a rich E. verrucosa forest as case study in the Atlantic Ocean can give a better comparison to the Mediterranean ones studied thus far. These missing information will be essential towards the development of a standardized and non-invasive method for the long-term monitoring of E. verrucosa and its population structure.”
Round 2
Reviewer 3 Report
Line 75 - remove '-enough'.
Line 75 - 'settling' not 'settled'.
Line 84 - '.....that makes recovery difficult after.....'
Line 131 - 'change' not 'changes'.
Line 132 - 'SW England', not 'SE England'.
Line 135 - after [27], put in a full stop.
Line 142 - 'a Citizen Science' (remove 'the').
Line 145 - remove 'an'.
Line 148 - remove the last 'the'.
Line 154 - poor English - remove 'for a potential application in a'
Line 244 'a' case study. (include the 'a')...
Line 260 - after 'Model A' put in a full stop, and start a new sentence with 'Hence'.
Line 292 - remove the first 'the'.
Line 304 - replace 'was' with 'were' at the end of the line.
Line 306 - replace 'afterwards' with 'in later years'.
remove line 365
Line 366 - remove 'the'
Line 402 - replace with 'were of a few large, old ones...'
Line 444 - remove 'this' and replace with 'the'.
Line 492 - poor English - remove 'indeed' and then put 'can allow for the identification and recording of vulnerable coral communities'.
line 496 - change 'fishery' for 'fishing gear'line 499 - poor English.
Line 499 - poor English - 'represents an opportunity to correlate.....'
Line 501 - poor English - 'represents a population' (not 'case study'). Put 'that' after 'Atlantic Ocean'.
Line 502 - poor English - replace 'these missing' with 'Such'...
Line 504 - poor English - re-write 'this could lead to visual methods.....' This is far too long a sentence. It needs to be split, as mentioned in the first comments.
Author Response
We warmely thank the reviewer for the detailed revision. We accepted all the suggestions. Please find here a point-by-point response.
Line 75 - remove '-enough'.
Reply: Done.
Line 75 - 'settling' not 'settled'.
Reply: Done.
Line 84 - '.....that makes recovery difficult after.....'
Reply: Done.
Line 131 - 'change' not 'changes'.
Reply: Done.
Line 132 - 'SW England', not 'SE England'.
Reply: Done.
Line 135 - after [27], put in a full stop.
Reply: Done.
Line 142 - 'a Citizen Science' (remove 'the').
Reply: Done
Line 145 - remove 'an'.
Reply: Done.
Line 148 - remove the last 'the'.
Reply: Done.
Line 154 - poor English - remove 'for a potential application in a'
Reply: Done.
Line 244 'a' case study. (include the 'a')...
Reply: Done.
Line 260 - after 'Model A' put in a full stop, and start a new sentence with 'Hence'.
Reply: Done.
Line 292 - remove the first 'the'.
Reply: Done.
Line 304 - replace 'was' with 'were' at the end of the line.
Reply: Done.
Line 306 - replace 'afterwards' with 'in later years'.
Reply: Done.
remove line 365
Reply: Done.
Line 366 - remove 'the'
Reply: Done.
Line 402 - replace with 'were of a few large, old ones...'
Reply: Done.
Line 444 - remove 'this' and replace with 'the'.
Reply: Done.
Line 492 - poor English - remove 'indeed' and then put 'can allow for the identification and recording of vulnerable coral communities'.
Reply: Done.
line 496 - change 'fishery' for 'fishing gear
Reply: Done.
Line 499 - poor English - 'represents an opportunity to correlate.....'
Reply: Done.
Line 501 - poor English - 'represents a population' (not 'case study'). Put 'that' after 'Atlantic Ocean'.
Reply: Done.
Line 502 - poor English - replace 'these missing' with 'Such'...
Reply: Done.
Line 504 - poor English - re-write 'this could lead to visual methods.....' This is far too long a sentence. It needs to be split, as mentioned in the first comments.
Reply: Done.